# Neuroplasticity within and between Functional Brain Networks in Mental Training Based on Long-Term Meditation

**DOI:** 10.3390/brainsci11081086

**Published:** 2021-08-18

**Authors:** Roberto Guidotti, Cosimo Del Gratta, Mauro Gianni Perrucci, Gian Luca Romani, Antonino Raffone

**Affiliations:** 1Department of Neuroscience, Imaging and Clinical Sciences, “Gabriele D’Annunzio” University Chieti-Pescara, 66100 Chieti, Italy; cosimo@unich.it (C.D.G.); maurogianni.perrucci@unich.it (M.G.P.); 2Institute for Advanced Biomedical Technologies, “Gabriele D’Annunzio” University Chieti-Pescara, 66100 Chieti, Italy; glromani@itab.unich.it; 3Department of Psychology, “La Sapienza” University Rome, 00185 Rome, Italy; antonino.raffone@uniroma1.it; 4School of Buddhist Studies, Philosophy and Comparative Religions, Nalanda University, Rajgir 803116, India

**Keywords:** functional connectivity, meditation, MVPA, brain networks, fMRI, neuroplasticity, machine learning

## Abstract

(1) The effects of intensive mental training based on meditation on the functional and structural organization of the human brain have been addressed by several neuroscientific studies. However, how large-scale connectivity patterns are affected by long-term practice of the main forms of meditation, Focused Attention (FA) and Open Monitoring (OM), as well as by aging, has not yet been elucidated. (2) Using functional Magnetic Resonance Imaging (fMRI) and multivariate pattern analysis, we investigated the impact of meditation expertise and age on functional connectivity patterns in large-scale brain networks during different meditation styles in long-term meditators. (3) The results show that fMRI connectivity patterns in multiple key brain networks can differentially predict the meditation expertise and age of long-term meditators. Expertise-predictive patterns are differently affected by FA and OM, while age-predictive patterns are not influenced by the meditation form. The FA meditation connectivity pattern modulated by expertise included nodes and connections implicated in focusing, sustaining and monitoring attention, while OM patterns included nodes associated with cognitive control and emotion regulation. (4) The study highlights a long-term effect of meditation practice on multivariate patterns of functional brain connectivity and suggests that meditation expertise is associated with specific neuroplastic changes in connectivity patterns within and between multiple brain networks.

## 1. Introduction

Meditation can be characterized as a set of mind–body practices involving the regulation of attention, awareness and mental states [1,2]. It can be usefully classified into two main styles—focused attention (FA) and open monitoring (OM) meditation—depending on how attentional processes and cognitive monitoring are set [1,3]. In the FA meditation style, attention is focused on a given meditation object in a sustained manner, with the implication of attention regulation processes preventing distraction (e.g., mind wandering) and refocusing on the meditation object [4]. Open Monitoring is based on non-reactive and non-judgmental monitoring of the contents of experiences and mind-body processes, primarily as a mean to become aware of emotional and cognitive patterns and to increase cognitive and emotional flexibility [5,6]. Meditation practices are increasingly relevant in society and have important clinical, educational and workplace applications [7].

The executive, attentional, and emotional regulatory processes involved in these two main forms of meditation may have a deep impact on the brain, on cognition, and on behavior [1,2,6,8]. Several studies have suggested that FA and OM meditation styles involve different brain regions and processes, in association with their different phenomenological, attentional, and cognitive features [1,3,9,10,11]. However, other studies have highlighted common neural processes involved in these meditation forms, plausibly in association with the common executive and meta-awareness functions of FA and OM meditation [2,6,12].

In the past years, it has been shown that brain processes cannot be functionally characterized using isolated brain activations, but rather in terms of large-scale networks that enable an effective information exchange between regions [13,14,15]. The characterization of large-scale brain networks appears crucial for the study of consciousness and meditation [6,9,16]. Moreover, FA and OM meditation forms may differentially modulate the activity of large-scale brain networks [8,9].

It can furthermore be hypothesized that FA and OM meditation expertise differentially affects neuroplasticity in terms of functional connectivity (FC) patterns in the brain, in association with the attentional, monitoring, and regulation processes involved in FA and OM meditation [1,9]. In particular, the recently proposed Brain Theory of Meditation [8] suggests that FA meditation leads to a sharpening of activations and coupling between Salience, Central Executive, and Default Mode networks, whereas OM meditation is characterized by a more distributed pattern of activations and higher coupling, in particular in the left hemisphere.

Another relevant subject of study involves the relationships between meditation expertise, age and neuroplasticity related to FA and OM meditation states and traits, since aging leads to decreased executive control, monitoring, and attentional performances [17,18], whereas meditation practices and expertise lead to enhanced and more efficient executive control, monitoring, and attentional processes [1,2,3,19].

The effects on connectivity have been addressed for both expertise [10,19,20] and age [21,22]; however, these investigations focused only on a subset of brain regions and networks and compared different groups of participants without directly assessing the specific modulation of brain FC by meditation expertise and age in FA and OM meditation forms, including patterns of brain networks related to the regulation of attention, emotion, and self [5].

A promising tool to investigate these effects is the study of patterns of FC through multivariate pattern analysis (MVPA). The interest in MVPA in neuroimaging [23,24] has grown rapidly due to the ability to detect fine-grained differences in neurophysiological patterns as compared with univariate methods [25]. Most MVPA studies have focused on cognitive and perceptual aspects of brain functions [26,27,28,29]. Recently, multivariate methods have been successfully used with FC, as a promising tool for the prediction of individual subject traits [30,31,32,33,34]. MVPA has also been used in meditation studies, using structural MRI to predict the age of a group of meditators [35] and discriminate between meditators and non-meditators [36] as well as to classify the effects on FC of a body–mind training course [37].

Here we studied the impact of meditation expertise and age on patterns of FC within and between large-scale brain networks in FA and OM meditation states. We sought to disentangle the modulation of meditation expertise and age on these networks by using pattern regression techniques. In particular, we assessed whether patterns of functional connectivity in FA and OM meditation states are predictive of the number of years of meditation practice (meditation expertise) and age in a group of long-term Theravada Buddhist monks, who extensively practice both forms of meditation.

Although meditation expertise and age are necessarily correlated, given their opposite effects on cognitive performance, it is likely that they affect functional connections in different ways. Moreover, given the different nature of FA and OM meditation practices, these should affect functional connections differently as well.

Specifically, we hypothesized that (i) the patterns of FC within and between functionally relevant brain networks in the two forms of meditation can be used to predict years of meditation practice (expertise) as well as age; (ii) age-modulated patterns are independent from the type of meditation; (iii) expertise-modulated patterns substantially differ between FA and OM meditation forms. In particular, we hypothesized that the connectivity pattern modulated by meditation expertise in FA meditation includes nodes and connections implicated in focusing, sustaining, and monitoring attention, while modulated FC patterns in OM meditation include nodes and connections associated with interoception, cognitive and affective monitoring, and regulation.

## 2. Materials and Methods

### 2.1. Participants

Twelve Theravada Buddhist monks (males, mean age 37.9 years, SD 9.4 years, range 25–60) from Santacittarama Buddhist Monastery in Central Italy, following the Thai Forest Tradition, participated in our study. Participants practiced FA (Samatha) and OM (Vipassana) meditation forms in a balanced way in this tradition, as suggested by the abbot of the monastery. Meditation expertise was measured in years of practice (mean = 16.41 years, SD = 7.69 years), with each year assumed to correspond to 1200 h of meditation practice. The experiment was conducted with the subjects’ written informed consent, according to the Declaration of Helsinki and with the approval of the Ethics Committee of “G. d’Annunzio” University of Chieti-Pescara.

### 2.2. Experimental Design

The experimental design consisted of three blocks of the following sequence: 6 min of FA and 6 min of OM meditation blocks intermixed with a 3 min non-meditative rest block (Figure 1A) and cued by vocal instructions. The total duration of the experiment was 57 min. Instructions on the meditation type or rest were provided verbally before the beginning of each block. Retrospective self-reports of the meditators suggested that they adequately performed both FA and OM forms during the experiment.

### 2.3. Data Acquisition and Preprocessing

Images were acquired on a 1.5 T Siemens Magnetom Vision Scanner; BOLD signal images were obtained using T2*-weighted echo planar (EPI) sequence (TR = 4.087 s, 28 slices, voxel size 4 × 4 × 4 mm^3^, 860 volumes).

A high-resolution T1-weighted whole-brain image was acquired with a 3D-MPRAGE sequence (sagittal matrix = 256 × 256, FOV = 256 mm, voxel size 1 × 1 × 1 mm^3^, flip angle = 12°, TR/TE = 9.7/4.0 ms).

Preprocessing of raw images was performed using Brain Voyager QX 1.7 software (Brain Innovation, Netherlands). The first five scans were discarded to reach T1 saturation. Preprocessed functional volumes were co-registered with the corresponding structural image. Temporal filtering included linear and non-linear (high-pass filter of two cycles per time course) trend removal. No spatial or temporal smoothing was applied, to avoid the increase of similarity of between-subject connectivity [38].

### 2.4. Functional Connectivity Analysis

We used the functional ROI parcellation proposed in Shirer et al. (2012) [39], which defines 90 ROIs from 14 large-scale brain networks. The selected ROI and network lists are presented in Appendix A.

We segmented anatomical images into grey matter, white matter, and cerebrospinal fluid (CSF) using the FSL FAST algorithm [40]. White-matter and CSF average signals were regressed out from the fMRI signal, together with movement parameters. Then, the residual signal was filtered (0.009–0.08 Hz) and finally scrubbed by removing the volumes with a Framewise Displacement (FD) index greater than 0.5 [41].

### 2.5. Registration and ROI Definition

We applied a nonlinear transformation to change the ROI coordinate space (MNI) into the subject space. This procedure is necessary since ROI coordinates were given in the MNI space; therefore a non-linear co-registration from the individual brain space to the MNI template was performed using the FSL FNIRT tool [42]. Then, the inverse of the obtained non-linear transformation was applied to the defined ROIs in order to extract the signal from the subject space. Once the inverse transformation was applied, we performed the intersection, in the individual space, between the ROI mask and the grey matter mask to obtain the ROI time courses. We selected only the ROIs that had an intersection with grey matter in all participants (7 ROIs discarded).

The average ROI time course was extracted, and the pairwise Pearson correlation coefficient was computed between all the ROIs, independently from the meditation form, and for each experimental block. Then, Fisher z-transformation was applied.

### 2.6. Multivariate Pattern Regression

The values from the upper triangle of the correlation matrix were used as features (Figure 1B). We computed a correlation matrix for each condition, block, and participant. Therefore, the dataset for pattern regression analysis was composed of six correlation matrices per participant (3 for FA and 3 for OM), resulting in 3403 features for 36 samples for each meditation form.

To reduce the difference between the number of features and the number of samples, we adopted a feature selection procedure: we computed Pearson correlation between each feature and meditation expertise or age, then we selected the first 70 connections with the highest absolute correlation as input to the regression algorithm [30]. The dataset was split into two parts: 75% of subjects were used as a training set and the remaining 25% as a testing set. To ensure the independence of data, we used in the training dataset only data from a set of subjects, and in the testing dataset, only data from the remaining set of subjects. This procedure was repeated 50 times, by randomly selecting subjects for the training set to provide a good estimator of the prediction error [43]. Feature selection was calculated in the training set and then was applied to the testing set [44].

We used the linear Support Vector Regression, with C = 1, and then we computed both the mean squared error (MSE) and the Pearson correlation (COR) between predicted and real values as metrics to estimate the prediction error.

The statistical significance was assessed through permutation tests [45] (n = 1000) to obtain the null distribution for MSE and COR. Bonferroni correction was used to assess for multiple comparisons.

The analyses were carried out using scikit-learn [46] and scipy/numpy [47] packages.

### 2.7. Relevant Feature Analysis

We extracted the weights of each connection tuned during the training phase. Since we selected only important features before training the prediction model, we discarded noisy features that could bias the interpretation of the model feature weights [48].

The weights were normalized using the z-score and averaged across cross-validation to obtain a single matrix with the contribution of each connection.

### 2.8. Control Analysis for Confound Effects

The two variables (age and expertise) could represent a reciprocal potential confound for the prediction task. The main approaches to control confounds are (i) using the confound as a feature to check its predictive power, but a good classification performance could be misinterpreted due to noise of the confound [48], and (ii) regressing out the confound from the data, but it may also remove some variance related to the variable of interest [49]. We performed a control analysis by regressing out the confound from the data (expertise for age prediction and vice versa).

Since we trained two separate predictive models, for age and expertise, we could also control for confounding effects by comparing the features selected by the two models [50]. If feature sets overlapped in the models, then the prediction was biased, while if the relevant feature sets were disjointed, the prediction was not affected by the confound variable. The advantage of this approach is that the target variable was directly predicted from the connectivity matrix, without removing the signal from a correlated confound as in regression, in order to better interpret the features used for prediction.

### 2.9. Control Analyses in a Novice Group

We performed an analysis by using the same model trained on the meditator group to predict the age of a novice group of subjects, aiming at understanding whether the trained model was generalizable or was specific for meditators.

The control group was composed of 10 novice meditators (males, mean age 33.0 years, SD 4.0 years, range 22–36), with no prior meditation expertise, recruited from the local community. They practiced FA and OM meditation for 10 days (30 min a day for each form of meditation) before taking part in the experiment. The specificity of the predictive model for meditators could strengthen our hypothesis about the long-term effects of meditation in FC modulation.

A separate analysis to understand whether age can be predicted by functional connectivity in FA and OM meditation forms in the control group was also performed.

Finally, we contrasted, by using a two-sided *t*-test, the average connectivities of meditators and novices, in FA and OM, to further validate our hypotheses.

## 3. Results

### 3.1. Prediction of Years of Meditation Expertise

The prediction performance was calculated using the mean squared error (MSE) and the correlation coefficient (COR). We predicted the years of meditation expertise with an MSE of 0.497 (*p* < 0.01) and a COR of 0.857 (*p* < 0.001) in FA meditation. During OM meditation we predicted the expertise with a MSE of 0.520 (*p* < 0.01) and a COR of 0.834 (*p* < 0.001). The average error of the model was about 0.71 years.

### 3.2. Prediction of Age

The age of the meditators could be predicted during the FA meditation (MSE = 0.186, *p* < 0.005; COR = 0.863, *p* < 0.001) and during OM meditation (MSE = 0.201, *p* < 0.005; COR = 0.791, *p* < 0.001). The average error was about 0.45 years for FA and OM.

### 3.3. Control Analyses for the Confounding Effect

Control analysis for the confounding effect was performed, since both variables were correlated (r = 0.68).

The results confirmed that we can predict expertise and age by regressing out the confounding variable (age and expertise, respectively). Specifically, the expertise can be predicted after regressing out the age effect in FA, with MSE = 0.531 (*p* < 0.01) and COR = 0.847 (*p* < 0.001), and in OM, with MSE = 0.563 (*p* < 0.01) and COR = 0.811.

The age could be predicted after regressing out the expertise effect as well, with MSE = 0.175 (*p* < 0.005) and COR = 0.875 (*p* < 0.001) during FA, while in OM we found MSE = 0.190 (*p* < 0.005) and COR = 0.813 (*p* < 0.001).

These results confirmed that prediction is not affected by confounding variables. The prediction error is generally worse in the expertise prediction than in the age prediction, due to some variance lost after regressing out the confounding variable.

### 3.4. Control Analyses in the Control Group

We used the models trained using meditator data to test the control group to understand whether the age predictive model can be generalized to a different group of subjects. The models did not generalize to the novice group in FA (MSE = 2.54, n.s.; COR = 0.04, n.s.) or in OM (MSE = 4.01, n.s.; COR = 0.2, n.s.) meditation.

The prediction of age using a model built on the control group gave no statistically significant results, neither in FA (MSE = 1.877, n.s.; COR = 0.14, n.s.) nor in OM (MSE = 3.59, n.s.; COR = −0.2, n.s.).

In addition, to add evidence that long-term practice affects functional connectivity, we contrasted the connectomes of meditators and control groups in both conditions. We used a two-sided *t*-test for the null hypothesis that connectivities of meditators and the control group have identical average values (see Appendix A). As mentioned above, the specificity of the predictive model for meditators and the differences in the connectivities between the two groups supported our hypothesis about the long-term effects of meditation in FC modulation, despite the fact that differences between the two groups can be ascribed to other factors (e.g., diet, educational level), which are difficult to control but are common in the vast majority of meditation studies [2].

### 3.5. Feature Selection Frequency

We plotted, in Figure 2 and Figure 3, the number of times a connection was selected by the feature selection algorithm for each cross-validation fold.

Figure 2 shows the sets of connections used to predict age and meditation expertise in FA (panel a) and in OM meditation (panel b). The figure suggests that the selection frequency of the features was uncorrelated for the two prediction tasks (FA: r = −0.05, *p* > 0.05; OM: r = −0.04, *p* > 0.05), since the dots lie mostly far from the diagonal. These results demonstrate that the connections that were involved in age prediction were different from those involved in predicting the meditation expertise.

In Figure 3 we plotted the frequency of the selected features for the prediction of meditation expertise (panel a) and age (panel b). Specifically, when features were used to predict the meditation expertise (panel a), feature selection frequencies appeared to largely differ in the two forms of meditation. In other words, the two meditation styles recruited different connections; this was confirmed by the low value (r = 0.14, *p* < 0.05) of the frequency correlation across features. Instead, when features were used to predict the age of the meditators (Figure 3b), the algorithm selected the same features in the two meditation styles, and the frequencies were highly correlated (r = 0.97, *p* < 0.001). This demonstrates that the connections affected by age are insensitive to the meditation style and are distinct from those involved in meditation.

We also observed the same effect when the confound variable was regressed out (r= 0.917, *p* < 0.001) (Appendix A).

### 3.6. Prediction-Relevant Feature Weight Analysis

Next, we analyzed the model weights to evaluate which connections and nodes were more predictive of meditation expertise and age in FA and OM meditation and to what extent. We then restricted the analysis to those connections that were selected at least in 95% of the trained models. For each connection, the weights were averaged across cross-validation folds and z-scored.

We averaged and normalized the connection weights within and between networks to have an integrated view of the contributions of each large-scale network, as shown in Figure 4, for the prediction of expertise in FA (panel a) and OM (panel b) meditation and for the prediction of age (panel c).

We further plotted, in Figure 5, the normalized absolute weights of the ten most important nodes for the prediction of meditation expertise (Figure 5a,b) and for the prediction of age in both meditation forms (Figure 5c,d).

In Figure 6 we plotted the contribution of the connections in the prediction of meditation expertise in FA (panel a) and OM (panel b) meditation.

In Figure 7 we show the connections that were used to predict expertise (panel a) and age (panel b) and those shared in both meditation styles; we chose those that were selected 95% of the time in the regression models.

## 4. Discussion

In this study, we investigated the modulation by meditation expertise and age on patterns of functional brain connectivity in brain networks during FA and OM meditation states.

We used pattern regression techniques to disentangle the influences of meditation expertise and age. We want to point out that the primary goal of this study was to leverage machine learning to investigate these effects and not to build a predictive model for meditation expertise and age.

The results show that meditation expertise and age can be predicted with significant accuracy from fMRI functional connectivity (FC) data recorded during meditation. The results also highlight a long-term effect of the extensive practice of meditation on FC in brain networks and shed light on the different contributions of meditation expertise and age on brain FC in meditation.

We found that expertise-based patterns differed between FA and OM meditation forms, while age-specific patterns were independent of the type of meditation. Specifically, the stability of the connectivity patterns modulated by age, across different meditation styles, suggests that the performed meditation moderately reorganizes age-specific connections, providing further evidence of the stability of brain networks [51]. The meditation-specific modulation of brain networks due to expertise demonstrates that long-term histories of co-activation among meditation-specific regions strongly impact the organization of brain networks [51], supporting the idea that connectivity profiles could predict cognitive behavior [31].

### 4.1. Prediction of Meditation Expertise in FA

In our investigation of the relevant features for prediction of expertise in FA meditation, we found an increase in connectivity associated with meditation expertise between the left intraparietal sulcus (IPS)–angular gyrus (AngG) and the anterior cingulate cortex (ACC), two brain regions that are involved in top-down attention and attentional control. In particular, the ACC is crucial for monitoring conflicts [52,53], such as the conflict between focus on the meditation object and distractions, while the IPS and AngG are directly involved in top-down attentional selection and sustained attention [4,54], two other important facets of FA meditation.

In addition, the white matter fractional anisotropy of ACC has been shown to increase in short-term meditators, suggesting an increased communication efficiency and an improvement of attention monitoring skills [55]. The increase of the correlation of the ACC with the IPS/AngG with an increase in expertise may reflect an efficient communication process for detecting distractions and reorienting attention, as in the proposed FA model by Hasenkamp et al. (2012) [4]. We also observed that FA meditation is associated with connections with a negative correlation with meditation expertise, including the connection between the posterior cingulate cortex (PCC) and the left occipital gyrus, which can be explained as a reduced occurrence of spontaneous thoughts and mental images during sustained attention in FA [11,56] and as the well-established involvement of PCC in mind wandering and stimulus-independent thought within the default mode network (DMN) [57,58].

We also found a positive correlation of expertise with the connectivity of nodes in the higher visual network, which is in line with studies that showed an increased involvement of higher visual areas in expert meditators [37,59,60,61]. The reduced expertise-modulated connectivity in nodes within the posterior salience network suggests that salient stimuli may be dropped in favor of a narrow focus of awareness on a single object [62].

### 4.2. Prediction of Meditation Expertise in OM

In the model proposed by Vago and Zeidan (2016) [62], the fronto-parietal network (FPN) and the ventral attention network play a key role in coordinating awareness and monitoring of experience in the present moment during OM meditation. Results showed that the dorsomedial PFC, which is a core region of the FPN, has a high correlation with the left Crus, a node implicated in emotional processing [63].

Our finding highlights the specific involvement of right caudate (RCau) in the OM facet of meditation (Figure 3b), which is characterized by a wider conscious access to the fields of experience. In addition, it has been demonstrated that expert meditators have greater widespread FC of the RCau with several brain regions [64]. Here we found an increased expertise-modulated connection of RCau with nodes in the Language Network, the left angular gyrus (LAG), and left middle temporal gyrus, which can be associated with an enhanced hub function of the caudate in multiple cognitive abilities [65].

Here we also found a negative connection weight between the right superior frontal gyrus and the right premotor cortex, which can be related to the reduced enactment (plausibly associated with the preparation of movement) of mental states (and in particular negative emotions) in OM meditation with increased meditation expertise [1].

Finally, a high correlation of connectivity between the left and right hippocampus with meditation expertise was shown, in line with the evidence of hippocampal involvement in mindfulness meditation [2,5]. Moreover, the hippocampus is a key convergence zone in the brain, integrating episodic information [66] and related emotional contents [1] as well as enhancing processes of interoception [67].

Our results highlighted a set of connections used to predict expertise in both OM and FA (Figure 7a). This shared pattern of connectivities may be related to common aspects of these two styles, since both forms involve a narrow focus on the present-moment experience and a detachment from reactivity and judgment patterns [62]; moreover, OM meditation is often practiced after FA meditation [1].

In our analysis, the model revealed a high implication of the posterior insula, which is involved in awareness of breath sensations [68,69] and is characterized by decreased connections with the right thalamus and the LAG, which may be associated with the prevention of broadcasting of thalamo-cortico-limbic signals associated with emotional reactivity, with a potential interference during FA and OM forms [16].

We also found an increased connectivity between the LAG and the middle frontal gyrus, which may be associated with the enhanced executive inhibition skills of meditators and plausibly related to changes in fronto-parietal FC [70,71,72].

Interestingly, our analyses showed the importance of regions in the left hemisphere in the prediction of expertise (Figure 6 and Figure 7). This finding can be associated with the hypothesis of leftward dominance of the brain in long-term meditators, which helps the top-down regulation of brain processes and states [8]. This hypothesis is supported by other findings by our group in the same set of subjects [10,11], in which the left brain hemisphere was involved in terms of activation and neural couplings in both forms of meditation. In addition, a meta-analysis and review highlighted the relevance of this hemisphere in long-term meditators [73]. Finally, studies in long-term meditators showed the prevalence of left regions for emotion processing [74,75] as well as structural changes related to attentive [35] and executive regions [76] in the left hemisphere.

### 4.3. Prediction of Age

The analysis revealed that the caudate-thalamus connections within the basal ganglia network have the highest importance for age prediction, which may be related with the importance of these regions in practitioners of yoga and insight meditation [64].

Our findings also revealed that thalamic connections and insular regions are central for the prediction of age from FA and OM meditation. This result appears consistent with the aging effects on the white matter changes observed in long-term meditators in these regions [77].

Further investigations should be performed to elucidate whether the age-specific patterns of FC are related to the reduction of neurocognitive decline with aging due to extensive meditation practice [78,79].

### 4.4. Functional Connectivity Patterns at the Brain Network Level

As shown in Figure 4, the analysis at the brain network level reveals interesting differential FC patterns related to expertise and age in FA and OM meditation forms.

In line with the analysis at the level of brain regions within and between networks, fewer expertise-modulated connections were found in FA meditation as compared to OM meditation.

We showed that functional connectivity in the posterior Salience Network (SN) was strongly modulated by meditation expertise in both FA and OM meditation forms, in terms of a reduced coupling, and by age (in both forms of meditation), in terms of an increased coupling. The posterior SN areas, such as the posterior insula, may play the role of key hubs in the regulation or modulation of sensory and feeling experiences, including pain feeling [80], plausibly in terms of down-regulation of activation and its coupling in meditative states [8,10,81]. The increased coupling within the posterior SN with age may reflect a reduced selectivity of neural responses related to sensory and feeling experiences of relevance in both FA and OM meditation forms. In OM meditation we also found an increased expertise-related connectivity between the posterior SN and the visuospatial network, which may be related to an enhanced conscious access to a range of sensory and feeling experiences during OM meditation [1,16].

In FA meditation, results demonstrated an increased expertise-related connectivity within the high visual network, as observed at the level of brain regions. This finding can be associated with a spontaneous focused visualization (with closed eyes) called *nimitta* in the Buddhist meditation tradition, which may arise during intense meditation [82].

The expertise-modulated connectivity pattern within and between brain networks in OM meditation also includes an increased coupling within the RECN as well as between this network and the language network. These enhanced functional connections may be related to a controlled conscious (metacognitive or mindful) access to sensory and thought contents arising during OM meditation, preventing judgments with affective valence as well as mind wandering [1,2,6,83].

Expertise-related modulation of functional connectivity in OM meditation also included the sensorimotor network, in terms of enhanced within-network connectivity, and reduced connectivity with the dorsal DMN. The first effect may be related to an increased conscious access to top-down regulation of perception and action during this form of meditation [1,16], while the second may be related to the functional segregation between (conscious) extrinsic sensorimotor processing and intrinsic mentation involving the dorsal DMN [4].

Finally, age-related modulation in both FA and OM forms implied a decreased coupling within the anterior SN, which may be explained by an age-related reduced efficiency in this network [84]. The increased FC between the basal ganglia and the auditory network may be explained by the effect of higher interference of auditory experiences, related to the fMRI-acoustic noise in our experiment. An increase in the striatal connectivity with the auditory cortex has also been found in tinnitus [85].

A limitation of our study is the small sample size used [86]. The vast majority of meditation studies involving long-term meditators used tens of subjects [2]; however, recently it has been found that a limited sample size can reveal the general principle of brain functioning more productively [87]. Several studies have used a comparable sample size for predictive analyses [37,88,89,90,91], also using highly non-linear models [92]. In addition, the robustness of the statistical analysis and the cross-validation schema [43] suggests that the conclusions are not over-inflated.

## 5. Conclusions

Our study demonstrated that fMRI connectivity patterns within and between brain networks can differentially predict meditation expertise and age of long-term meditators, with different patterns in FA and OM meditation forms. Using machine learning techniques, we showed that the connectivity patterns that predicted meditation expertise depended on the meditation style, while age-predictive patterns were the same for both meditation styles. In conclusion, these findings suggest that connectivity patterns in brain networks are differentially shaped by meditation expertise and age, with differential influences of focused attention and open monitoring meditation forms, confirming the specificity of the mental (cognitive and emotion regulation) processes involved in these main forms of meditation [12].

## Figures and Tables

**Figure 1 brainsci-11-01086-f001:**
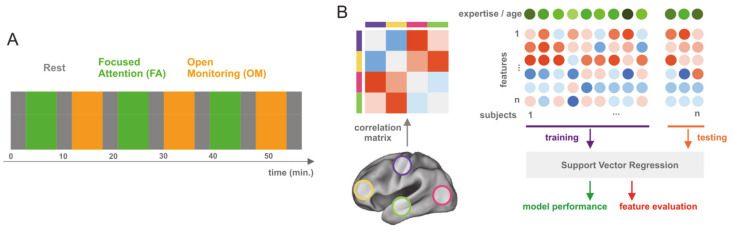
Experimental paradigm and analysis schema. (**A**) Experimental procedure consisted of three blocks of the following sequence: 6 min FA and 6 min OM meditation blocks intermixed with 3 min non-meditative resting state block. (**B)** Analysis schema consisted in extracting the average time course of the preprocessed BOLD signal (see Section 2) from 90 ROIs and computing the pairwise Pearson correlation matrix between extracted time courses; then, the upper triangular part of the correlation matrix was used as a feature set for training a Support Vector Regression (SVR) model, to predict either the meditation expertise or the age of the participants. Before training, a correlation based feature selection was performed to reduce the feature set. We split the dataset into two parts: 75% of the subjects were used for training and 25% for training repeating this procedure 50 times, with permutation of the set of subjects for cross-validation. Finally, performance was evaluated using mean squared error and correlation, while feature evaluation was performed by extracting the selection frequency of the features and by inspecting the weights of the SVR model.

**Figure 2 brainsci-11-01086-f002:**
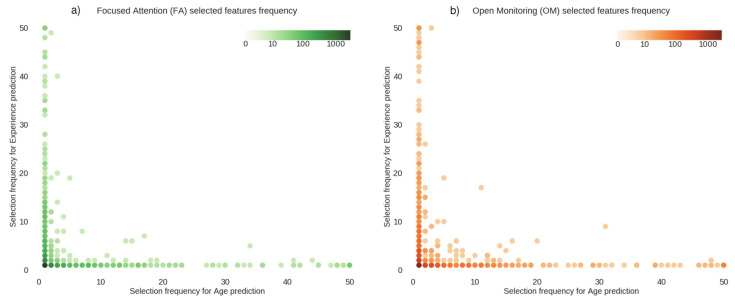
Feature selection frequency in FA and OM meditation styles. The figure shows the feature selection frequency for (**a**) focused attention and (**b**) open monitoring meditation, for age prediction (*x*-axis) and expertise prediction (*y*-axis). Each dot represents a connection (feature), and the position coordinates x and y represent the number of times this connection was selected in the regression, in all cross-validation folds. The maximum frequency value is equal to the number of cross-validation folds. The color of the dots indicates the number of features with that particular selection frequency. In these plots, dots representing features with similar frequencies in the two regression tasks lie close to the diagonal, while dots representing features with different frequencies lie far from the diagonal.

**Figure 3 brainsci-11-01086-f003:**
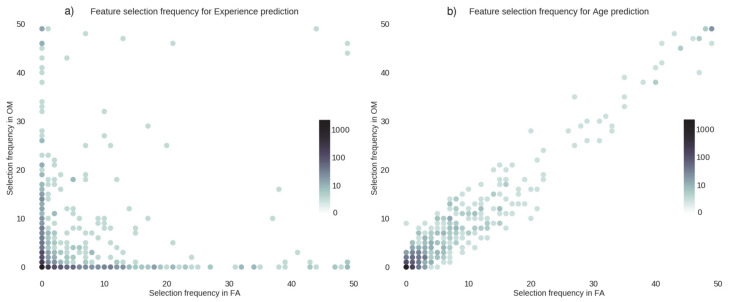
Feature selection frequency for expertise and age prediction. The figure shows feature selection frequency for (**a**) meditation expertise prediction and (**b**) age prediction for focused attention (*x*-axis) and open monitoring (*y*-axis) meditation styles. Each dot represents a connection (feature), and the position coordinates x and y represent the number of times this connection was selected in the regression, in all cross-validation folds. The maximum frequency value is equal to the number of cross-validation folds. The color of the dots indicates the number of features with that particular selection frequency. In these plots, dots representing features with similar frequencies in the two meditation styles lie close to the diagonal, while dots representing features with different frequencies lie far from the diagonal.

**Figure 4 brainsci-11-01086-f004:**
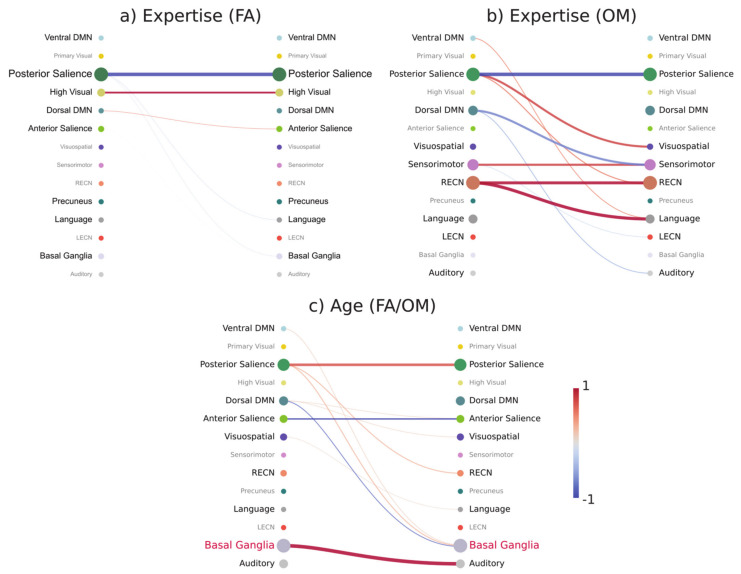
Within− and between−network contributions for different prediction tasks. (**a**) The average of connectivity weights within and between networks for expertise prediction in FA meditation style and (**b**) OM style and (**c**) for age prediction in both styles. These weights were obtained by extracting the weights learned by the model, averaging within and between networks, then normalizing them using the z-score function. Straight lines represent within-network contributions, while other lines represent between-network contributions.

**Figure 5 brainsci-11-01086-f005:**
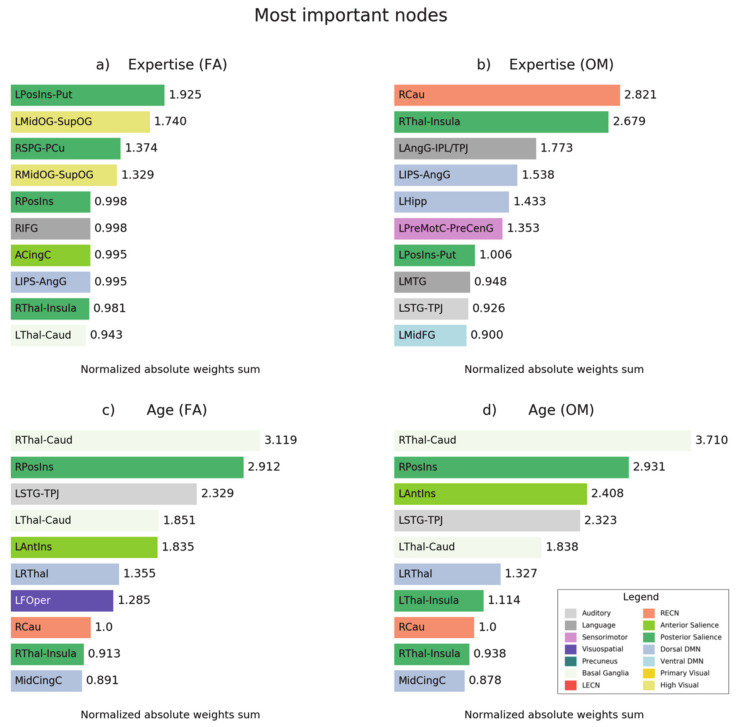
Normalized absolute weights of the nodes for different prediction tasks. The figure shows normalized absolute weights for (**a**) expertise prediction in FA meditation style and (**b**) OM style and (**c**) for age prediction in FA style and (**d**) OM style. These weights were obtained by extracting the absolute weights learned by the model, averaging across folds, and then normalizing to have mean equal to 1 and standard deviation equal to 0. Finally, for each node, we averaged the weights between connections that included that particular node.

**Figure 6 brainsci-11-01086-f006:**
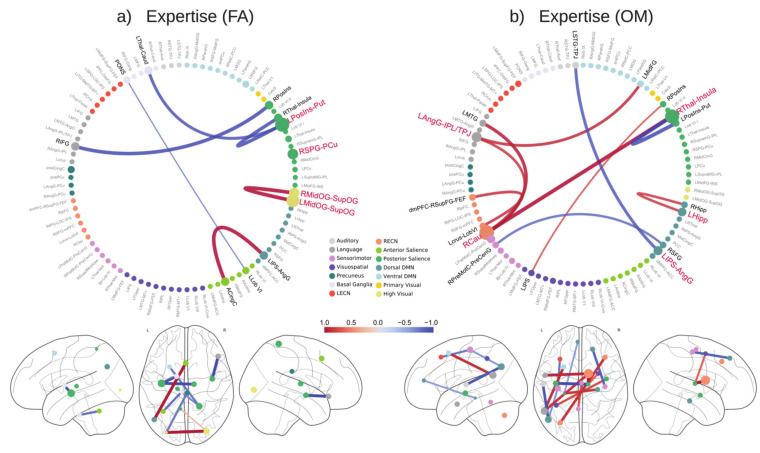
Meditation−specific weights of the expertise prediction model. The figure shows the normalized feature weights of the expertise prediction model averaged across cross-validation folds. These weights are specific for expertise prediction during (**a**) FA and (**b**) OM. A positive (red) weight indicates an increase of connectivity between nodes as the expertise increases, while a negative (blue) weight indicates a decrease of connectivity as the expertise increases. The circle size shows the absolute average weight of connections that include that specific node.

**Figure 7 brainsci-11-01086-f007:**
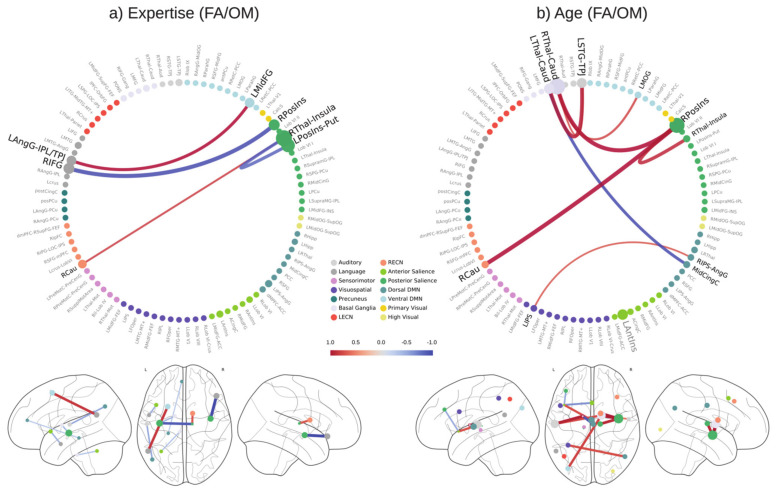
Connectivity weights shared across different meditation styles. The figure shows the normalized feature weights of the predictive connectivities that are shared in the meditation styles. Weights were averaged across cross-validation folds. These weights were learned by the model to predict (**a**) expertise and (**b**) age in both FA and OM meditation. The shown connectivities are those chosen by the feature selection algorithm in at least 95% of the folds in both FA and OM meditation. A positive (red) weight indicates an increase of connectivity between nodes as the expertise increases, while a negative (blue) weight indicates a decrease of connectivity as the expertise increases. The circle size shows the absolute average weight of the connections that include that specific node.

## Data Availability

The datasets generated and/or analyzed during the current study are available from the corresponding author on request.

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
