# Peer review of "Neuroplasticity within and between Functional Brain Networks in Mental Training Based on Long-Term Meditation"

_brainsci, 2021, doi:10.3390/brainsci11081086_

Round 1

Reviewer 1 Report

The study describes the effects of two types of meditation on patterns of functional connectivity in long-term meditators. The authors show that functional connectivity can predict meditation expertise and age, and that expertise predictive patterns are specific for the type of meditation while age predictive patterns are rather general. In general, this is very good work. The paper reads well, the results are presented both in written and graphic form and are easy to grasp, the discussion of existing literature is sufficient. I have only minor remarks to improve the Ms.

Minor points:

  1. The participants practiced focused attention Samatha meditation. It would be good to clarify if the breath was used as a focus. It would help to relate this type of meditation with the finding of the involvement of the posterior insula, which is implicated in the awareness of breath.
  2. In section 2.5, please clarify how 36 samples were obtained from 6 meditation blocks in each participant.
  3. Figure 4, caption: last sentence is not complete.
  4. Figures 6 and 7 suggest that patterns in the left hemisphere were more predictive for expertise and age than patterns in the right side of the brain. Could the authors comment if the results indeed suggest that long-term effect of meditation might be associated with asymmetry of brain activity?

Reviewer 2 Report

The study aimed to learn functional connectivity patterns associated with FA and OM. Its main novelty of the study is the applied multivariate pattern analysis. However, the details regarding the analysis method are missing. For instance, The analyses were carried out using scikit-learn [45] and scipy/numpy [46] packages.

I have two major concerns about the methodology of the paper.

  • The paper uses pair-wise functional connections, which are 3403 features. It seems that the authors are applying support vector regression on all 3403 features, which will have the overfitting issue (the curse of dimension), especially for such a small sample size. Therefore, the feature learned may not reflect the realistic association of functional connectivity with meditation experience (or age). You may want to focus a small number of features by feature selection from the current features or using network-based functional connections (e.g., 10 networks).

  • You do not have another data set to validate the features that you extracted. You have a control data set, which showed that your model cannot be generalize to. It cannot be determined whether your model is not good or the control data is fundamentally different from long-term meditation. You may compare the functional connectivity differences between 10-day meditation group and long-term meditators.

 A few minor concerns are listed below.

  • All the figures are very low resolution and it is hard to assess the contents.
  • Line 130: you mean 25% for testing?
  • Line 164: Please explain what is FD. It is not clear how the nonlinear transformation was calculated. It is better if you put the ROI derivation in the subject space in the preprocessing “registration” paragraph.
  • Line 175: It is not clear why you had 36 samples. Are they from FA, OM, resting-state three status for each of 12 subjects? But it seems that you did not use resting-state data in the paper.
  • Line 182: not sure what you mean for shuffling subjects in the training set? You only change the order of subjects in your pre-select training data set?
  • Line 188: Please clarify whether you really used Bonferroni correction? How many multiple comparisons did you have? 1000 permutations?
  • Line 224: How did the novice meditators learn FA and OM? By following a CD?
  • Line 467: a typo for “Posterion”.

Round 2

Reviewer 2 Report

The authors addressed most of my comments. I still think that the FC comparison graph between novice meditators and experts (shown in the Response letter) is important and it should be included in the manuscript. It is fine if you want to show it in the supplements. In addition, you want to discuss how the results are related to the features from your expertise-predictive model.

Author Response

The authors addressed most of my comments. I still think that the FC comparison graph between novice meditators and experts (shown in the Response letter) is important and it should be included in the manuscript. It is fine if you want to show it in the supplements. In addition, you want to discuss how the results are related to the features from your expertise-predictive model.

R: We thank the reviewer for the positive comments. We added the figure in the Supplementary Material, as proposed by the reviewer. Indeed, the contrast between expert and novice connectomes can add evidence to the main conclusions of our work, despite this analysis being out of the main scope of the paper.

As already mentioned in the previous response, discussing the differences between the features found in the predictive model and in the contrast between novice and meditators can be problematic due to the different nature of analyses and goals, and can be biased by several factors that are difficult to control (Tang et al., 2015). We also added this comment in the manuscript.

We found no substantial overlap between the features found by the prediction algorithm and the group comparison, but this is not surprising for several reasons. For example, the predictive model uses the joint information contained in a pattern of connectivities to predict the expertise of the meditator, while the t-test performs a comparison of the average of a single connection (O’Toole et al., 2007).  

In addition, the t-test is able to find the connections with different means, but is sensitive also to small variance distributions, while in the prediction task we are finding those connections in which the variance is large enough to explain differences in the level of expertise of the long-term meditators. Furthermore, we did not expect that the control group has predictive-based connectivities with a higher (or lower) average correlation value than meditators, but that long-histories of co-activation during meditation could modulate the connectivity strength. 

Another possible motivation is that the model is trained using a specific population of subjects, in which it behaves well, while a generalization to another completely different population should be carefully performed since it could not be appropriate and may bias the interpretation of the results (Hasson et al., 2020). 

Furthermore, there could be a set of confounding variables that can potentially harm the comparison, indeed, although novice performed 10 days of training, the differences may be due to other variables in the two groups and not accountable to the extensive meditation practice (e.g. education level, monastic life, diet, personality, attitude, meditation performance) (Tang et al., 2015). 

From a neuroscientific perspective, our results are in line with the study of Brewer et al. (2011), in which they found a stronger connectivity in the meditators group, compared to novices, in areas within the DMN, which have been addressed to a lower engagement of mind-wandering processes. In addition, the study of Hasenkamp et al. (2012) further validates our results, since they found higher connectivity between nodes in the executive network and in the salience network. In addition, we found some overlaps in nodes involved in the prediction of expertise and nodes in meditators versus novice contrast, for example in FA the posterior Insula and the Thalamus are clearly involved in the meditation due to their role in emotion processing and breath awareness (e.g. Marchand 2015, Kuehn et al. 2016). We can suppose that the overall differences between novice and meditators can be addressed to coarse difference in these two groups due to how easily these groups perform basic meditation task (e.g. conflict monitoring, attention orientation), while fine-grained modulation of connectivity due to extensive and long-lasting meditation can be catched by the use of multivariate pattern analysis within the same group of subjects.

In conclusion, we think that cross-validation and permutation tests demonstrated that the model can generalize well in out-of-sample meditators and the prediction is driven by the target variable, and not happening by chance, respectively. Nevertheless, we agree that to effectively validate our hypothesis a longitudinal study would have been more appropriate, but finding a trade off between a cross-sectional and longitudinal studies, especially in this “specialized” group of subjects, is not practical and common to the vast majority of meditation studies (Tang et al., 2015, Naselaris et al. 2021).

References:

  • Tang, Y.-Y., Hölzel, B.K., Posner, M.I.: The neuroscience of mindfulness meditation. Nat. Rev. Neurosci. 16, 213–225 (2015). https://doi.org/10.1038/nrn3916
  • Hasenkamp, W., Barsalou, L.W.: Effects of Meditation Experience on Functional Connectivity of Distributed Brain Networks. Front. Hum. Neurosci. 6, 38 (2012). https://doi.org/10.3389/fnhum.2012.00038
  • Brewer, J.A., Worhunsky, P.D., Gray, J.R., Tang, Y.-Y., Weber, J., Kober, H., 2011. Meditation experience is associated with differences in default mode network activity and connectivity. Proc. Natl. Acad. Sci. U. S. A. 108, 20254–20259. https://doi.org/10.1073/pnas.1112029108
  • Marchand, W.R., 2014. Neural mechanisms of mindfulness and meditation: Evidence from neuroimaging studies. World J. Radiol. 6, 471–479. https://doi.org/10.4329/wjr.v6.i7.471
  • Kuehn, E., Mueller, K., Lohmann, G., Schuetz-Bosbach, S., 2016. Interoceptive awareness changes the posterior insula functional connectivity profile. Brain Struct. Funct. 221, 1555–1571. https://doi.org/10.1007/s00429-015-0989-8
  • Hasson, U., Nastase, S.A., Goldstein, A., 2020. Direct Fit to Nature: An Evolutionary Perspective on Biological and Artificial Neural Networks. Neuron 105, 416–434. https://doi.org/10.1016/J.NEURON.2019.12.002
  • O’Toole, A.J., Jiang, F., Abdi, H., Pénard, N., Dunlop, J.P., Parent, M.A., 2007. Theoretical, statistical, and practical perspectives on pattern-based classification approaches to the analysis of functional neuroimaging data. J. Cogn. Neurosci. 19, 1735–1752. https://doi.org/10.1162/jocn.2007.19.11.1735
  • Naselaris, T., Allen, E., Kay, K., 2021. Extensive sampling for complete models of individual brains. Curr. Op. Behav. Sci. 40, 45-51. https://doi.org/10.1016/j.cobeha.2020.12.008